# IS THE REVERSAL CURSE A BINDING PROBLEM? UNCOVERING LIMITATIONS OF TRANSFORMERS FROM A BASIC GENERALIZATION FAILURE

**Boshi Wang**
The Ohio State University
wang.13930@osu.edu

**Huan Sun**
The Ohio State University
sun.397@osu.edu

## ABSTRACT

Despite their impressive capabilities, LLMs exhibit a basic generalization failure known as the *Reversal Curse*, where they struggle to learn reversible factual associations. Understanding why this occurs could help identify weaknesses in current models and advance their generalization and robustness. In this paper, we conjecture that the Reversal Curse in LLMs is a manifestation of the long-standing *binding problem* in cognitive science, neuroscience and AI. Specifically, we hypothesize two primary causes of the Reversal Curse stemming from transformers' limitations in conceptual binding: the *inconsistency* and *entanglements* of concept representations. We perform a series of experiments that support these conjectures. Our exploration leads to a model design based on JEPA (Joint-Embedding Predictive Architecture) that for the first time breaks the Reversal Curse without side-stepping it with specialized data augmentation or non-causal masking, and moreover, generalization could be further improved by incorporating special memory layers that support disentangled concept representations. Our research opens up the broader fundamental challenge of designing models capable of learning systematic conceptual binding with less human scaffolding.[1]

## 1 INTRODUCTION

Current large language models (LLMs) exhibit a notable failure of basic generalization known as the *Reversal Curse* (Berglund et al., 2024), where they struggle to learn rules of inversion over parametric knowledge and form reversible factual associations. For instance, after internalizing the fact *"Tom Smith's wife is Mary Stone"*, LLMs fail badly at recalling *"Tom Smith"* when asked *"Mary Stone's husband is __"*.[2] Reversal is not confined to natural language; it represents a class of basic operations across various domains such as mathematics/logic and numerous scientific disciplines, where inverse relationships are commonplace. Given that LLMs are trained on web-scale corpora containing data more than enough for inducing these rules, it is clear that there are missing inductive biases in current transformer-based language models (Vaswani et al., 2017; Brown et al., 2020; Chowdhery et al., 2023; Touvron et al., 2023) that hinder this kind of generalization. The simplicity of the rules also suggests their limitations in learning more complex skills and principles, which could hurt both their general abilities and potential to specialize into domain experts.

There are several pieces of work trying to understand or mitigate the Reversal Curse. Zhu et al. (2024) theoretically shows that reversal cannot be learned for transformers under special settings and assumptions. Lin et al. (2024) shows that the issue may be related to inherent biases in LLMs' factual recall. Golovneva et al. (2024); Guo et al. (2024); Lu et al. (2024); Lv et al. (2024); Kitouni et al. (2024) propose specialized data augmentation strategies (e.g., reversing/permuting sentence segments) or non-causal training objectives, which circumvent the problem and reduce generalization demands on models. Overall, existing solutions are ad-hoc and fall short of uncovering the potentially more foundational issues behind such a curse—in fact, the very first question still remains a mystery:

---

[1]Code and data: https://github.com/OSU-NLP-Group/reversal-curse-binding.

[2]A special case is when the involved relations are symmetric, which leads to examples like *"A is B"* then *"B is A"*, a popular reference to the Reversal Curse.

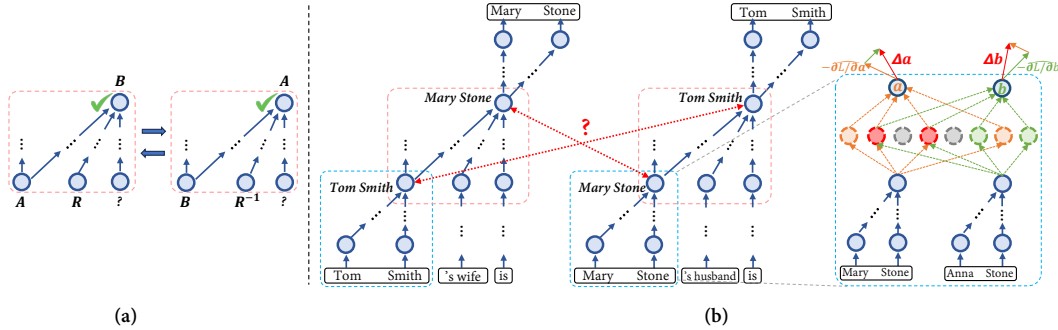

Figure 1: **(a)** We find that Transformers can learn reversal when inputs are represented and perceived at the *abstract concept level*. **(b)** Two conjectured causes of the Reversal Curse underlying surface-level predictions, both upon transformers' limitations in *conceptual binding*: 1) representational inconsistency when entities switch roles between perceived subjects and predicted objects (**left**); 2) representational entanglements cause interferences on the learning dynamics and impede generalization (**right**). Details in §2 and §3.

*Are conventional (autoregressive) transformers fundamentally doomed for learning reversal?*

Surprisingly, the answer is *"No"*. Our first major finding is that *standard transformers can learn reversal without any specialized data augmentation or modifications to the architecture or objective*, when the inputs are represented and perceived at the *abstract concept level* (Figure 1(a)). We then focus on the gap between abstract and real settings, where inputs are instead at the surface form level. Our investigations lead us to hypothesize that the Reversal Curse is fundamentally a manifestation of the long-standing *binding problem* in cognitive science, neuroscience and AI, which is concerned with the mechanisms for natural or artificial neural networks to combine information distributed throughout the network to form integrated percepts and knowledge (Roskies, 1999; Engel & Singer, 2001; Zimmer et al., 2006; Greff et al., 2020). Specifically, we conjecture that the Reversal Curse is primarily caused by two limitations of *conceptual binding* in transformers, the *inconsistency* and *entanglements* of concept representations:

- **Inconsistency.** While many existing studies show that LLMs could form internal concepts and even "world models" from surface-level predictions (Meng et al., 2022; Geva et al., 2023; Lad et al., 2024; Kaplan et al., 2025; Li et al., 2023; Gurnee & Tegmark, 2024), we hypothesize that they are still unable to adequately learn *consistent* concept representations across various places within the network under different contexts. Specific to reversal, we conjecture that transformers fail to *bind representations of the same underlying entity when it switches roles between perceived subjects and predicted objects* (Figure 1(b), Left), which makes the model's acquired knowledge fragmented and impedes the learning of reversal.
- **Entanglements.** Since concepts are activations in transformers, their representations can only be *indirectly* updated by altering the lower-level weights in the recognition module mapping surface-form names to concepts. We conjecture that transformers with gradient-based optimizations face difficulties in *maintaining the separation of distinct concepts during learning due to representational entanglements* (Figure 1(b), Right), which impacts the training dynamics and hinders generalization.

A series of quantitative experiments support our hypotheses, and inform two novel designs for mitigating the Reversal Curse: 1) performing autoregressive prediction at the concept level, akin to Joint-Embedding Predictive Architectures (JEPA) (LeCun, 2022) and concept models (Barrault et al., 2024), and 2) building dedicated recognition modules which support disentangled concept representations. We show that 1) a model design based on JEPA and in-batch contrastive learning could, for the first time to our knowledge, break the Reversal Curse with non-trivial performance without circumventing the problem, but suffers from entanglements that scale with model depth; 2) incorporating special memory layers (Sukhbaatar et al., 2015; Berges et al., 2024) into the recognition module could further boost generalization. We also demonstrate that the reversal skill unlocks a new kind of parametric memory integration, which allows models to perform *parametric forward-chaining*

| | $|\mathcal{E}_A| = 2.5K$ | $|\mathcal{E}_A| = 10K$ | $|\mathcal{E}_A| = 50K$ | $|\mathcal{E}_A| = 100K$ |
|---|---|---|---|---|
| #Layer = 1 | 0.823 | 0.861 | 0.947 | 0.964 |
| #Layer = 6 | 0.890 | 0.858 | 0.951 | 0.861 |
| #Layer = 12 | 0.810 | 0.878 | 0.951 | 0.960 |
| #Layer = 18 | 0.823 | 0.850 | 0.944 | 0.975 |

Table 1: Mean reciprocal rank (MRR) achieved by standard transformers in the abstract setting, where inputs are represented at the concept level. Here $\mathcal{E}_A$ is the set of entities in the training set for learning/inducing the rules of reversal.

during information internalization to solve large-scale arithmetic reasoning problems with impressive performance, outperforming frontier LLMs based on non-parametric memory.

To summarize, our work 1) contributes towards understanding and addressing the Reversal Curse, and more importantly, 2) connects the Reversal Curse with the more foundational problem of *improving conceptual binding and generalization* in AI models, rigorously establishing the concrete challenges for the broader research community.

## 2 LEARNING REVERSAL AT THE CONCEPT LEVEL

Humans think and learn at the concept level. When reading a sentence, we (usually subconsciously) parse and map the words into concepts, and update the concept representations and associations upon encountering new information (Collins & Loftus, 1975; Jackendoff, 1995). While transformers fail to learn reversal in real settings, do they first have appropriate inductive biases to learn reversal at the abstract concept level?

**Concepts in reversal.** Reversal is a simple and clean task involved with some of the most basic low-level concepts: entities and relations. Take *"Tom Smith's wife is Mary Stone"* as an example: each fact consists of the subject entity (*"Tom Smith"*), relation (*"'s wife"*), and the object entity (*"Mary Stone"*). At the concept level, each fact is hence $(e_1, r, e_2)$, and its reverse is $(e_2, r^{-1}, e_1)$ which contains the same piece of information. If a model acquires reversal, then after internalizing a certain fact in one direction, i.e., its parameters are changed s.t. $p(e_2|e_1, r, ?)$ is large, the model should also assign a high probability $p(e_1|e_2, r^{-1}, ?)$ to its reverse direction.

**Setup.** We prepare a set of relation pairs $\{(r_i, r_i^{-1}) \mid i = 1, \dots, N\}$, and two disjoint sets of entities $\mathcal{E}_A$ (for learning) and $\mathcal{E}_B$ (for testing). We focus on one-to-one relations that give a unique object entity for each fact. We synthesize facts separately over $\mathcal{E}_A$ and $\mathcal{E}_B$ by randomly pairing the entities for each $(r_i, r_i^{-1})$, and form a pair of facts which are reverses of each other over each entity pair. Through this, we obtain two sets of facts $D_A$ and $D_B$ over $\mathcal{E}_A$ and $\mathcal{E}_B$ respectively. The training data contains all of $D_A$ (for the model to induce the rules) and one random direction from each pair of facts in $D_B$, where its reverse goes into the test set. We set $N = 6$, and vary $|\mathcal{E}_A|$ while keeping the ratio $|\mathcal{E}_A| : |\mathcal{E}_B| = 5 : 3$. Importantly, *each concept (entity/relation) is directly represented by its own learnable embedding*, without attaching to surface-level names. We train standard decoder-only transformers as in GPT-2 (Radford et al., 2019) (with 768 hidden dimensions and 12 attention heads) to predict the object entity in each fact, with cross-entropy loss over embeddings of all concepts. We train models for a large number of steps ($3e6$) and report the highest mean reciprocal rank (MRR) on the test examples achieved among different model checkpoints. More training details are included in Appendix A.

**Transformers can learn reversal at the concept level.** Results are in Table 1. Surprisingly, in contrast with the negative results and views in previous studies, we find that *transformers can learn reversal with high performance without any specialized training objectives or data augmentation*. Models with different depths could all strongly generalize, where the performance overall increases with $|\mathcal{E}_A|$. Despite being extremely straightforward, this positive result becomes one of the major findings in this work. The critical question then arises: *If transformers can learn reversal at the abstract concept level, why do they fail in realistic settings?*

## 3   The Binding Problem Underlying Surface-level Predictions

The main difference in realistic settings is that the model perceives surface-form names instead of processing concepts directly, where the "curse" somehow arises. In this section, we analyze the main challenges of learning reversal through surface-level predictions, accompanied with quantitative experiments supporting our conjectures.

**The binding problem.** Our central thesis is that the Reversal Curse is a manifestation of the long-standing *binding problem* in cognitive science, neuroscience and AI, which is concerned with the mechanisms for natural or artificial neural networks to *bind information distributed throughout the network and form integrated percepts and knowledge* (Roskies, 1999; Engel & Singer, 2001; Zimmer et al., 2006; Greff et al., 2020). The binding problem could be divided into two major types: **perceptual binding** and **conceptual binding**. *Perceptual binding* refers to the combination of features from raw inputs into cohesive concepts, typically occurring during low-level perception/recognition (Von Der Malsburg, 1994; Tallon-Baudry & Bertrand, 1999; Singer, 2007; Palmigiano et al., 2017). *Conceptual binding* is centered around forming unified and integrated long-term knowledge, which occurs during high-level semantic processing and memory consolidation (McNorgan et al., 2011; Opitz, 2010; Murre et al., 2006; Patterson et al., 2007; Ralph et al., 2017).

Numerous studies show that LLMs have no trouble learning perceptual binding through surface-level predictions. For instance, prior work finds that detokenization is typically carried out in a "recognition module" within the lower layers, where subword tokens are combined and mapped into cohesive concept representations at the end of surface names (Meng et al., 2022; Geva et al., 2023; Lad et al., 2024; Yang et al., 2024; Kaplan et al., 2025); in upper layers, tokens of the output surface name beyond the immediate next token are also usually (often-times, linearly) encoded in the hidden states (Pal et al., 2023; Belrose et al., 2023; Wu et al., 2024; Cai et al., 2024). Complementary studies tracing the evolvement of LLMs' internal states during inference suggest that, beyond perceptual binding, they also "think" in an abstract concept space within a "semantic module" in the middle layers (Geva et al., 2022; Wendler et al., 2024; Lad et al., 2024; Sun et al., 2025). An illustration is in Figure 1(b). These findings indicate that the issues lie not in perceptual binding, but in the specific *representations* learned under surface-level prediction. Upon closer examination, we identify two key potential factors contributing to the failure in learning reversal, both stemming from transformers' deficiency in conceptual binding: **inconsistency** and **entanglements**.

### 3.1   Inconsistency of Concept Representations

We conjecture that one major cause of the Reversal Curse is that transformers lack inductive biases to learn *consistent* concept representations when they emerge across different contexts. This skill is performed seamlessly by humans; for example, when separately reading *"the city that held the 2024 Summer Olympics"* and *"the center of political change during the French Revolution"*, despite activating from and carrying different contexts (sports and history), the representations formed in our mind are bound and connect to the same underlying concept *"Paris"* instead of being isolated from each other.[3] Concretely for reversal, inconsistency instantiates into the failure of *binding entity representations when they switch roles between the perceived subjects and the predicted objects*, which emerge at the lower and upper layers within the model respectively (Figure 1(b), Left). Due to this, facts that are reverses of each other cannot be well integrated as one piece, impeding the induction of reversal.

Conceptual consistency is challenging to learn well within the design of current transformers even with an abundance of data, due to the dynamic and open-ended nature of concepts. Firstly, concepts in transformers could emerge at various locations/subspaces, necessitating a mechanism for dynamically tracking and connecting the concept representations across different regions of the network. Secondly, concepts are continuously assimilated instead of coming from a fixed vocabulary, which requires a systematic design that can establish the binding of newly acquired concepts automatically. This level of systematicity is not achieved even for the much simpler problem of binding tokens at the

---

[3]Consistency of concept representations is also the central thesis of the *Hub-and-Spoke* model, a prominent framework for human semantic memory (Patterson et al., 2007; Ralph et al., 2017) supported by evidence from neuroanatomy and studies on memory-impaired patients, which proposes that different experiences bind through a shared central 'hub' storing core concept representations, allowing knowledge integration and conceptual generalization.

input/output levels, especially for models with untied input/output embeddings (which is common in most LLMs today)—when *new* tokens are introduced to the vocabulary, transformers still have to rely on dedicated data to establish their binding.

## 3.2 ENTANGLEMENTS OF CONCEPT REPRESENTATIONS

Whereas consistency is involved with *connecting* representations of the *same* concept, problems also arise on the other side of the same coin—*separating* representations of *distinct* concepts. We conjecture that transformers lack inductive biases to decouple abstract mental concepts from direct perceptions during learning, an issue which we call *entanglement*. The inability to maintain the separation of distinct concepts could influence the training dynamics and negatively affect generalization.

To illustrate, consider the last MLP layer in the recognition module before concept representations are formed, as shown in Figure 1(b), Right. Suppose we have two activated concepts $a$ and $b$ in the current learning step, which have MLP hidden activations $\alpha, \beta$ respectively. Let the output projection matrix be $V$ where $v_i$ is its $i$-th column, and hence $a = \sum_i \alpha_i v_i, b = \sum_i \beta_i v_i$.[4] During learning when the loss is $L$, the negative gradients $-\partial L/\partial a$ and $-\partial L/\partial b$ represent the "desired directions" for updating $a$ and $b$. Assuming for now that $\alpha, \beta$ remain constant, we could compute the updates of concept representations $\Delta a$ and $\Delta b$ after a gradient descent step with step size $\eta$:

$$\frac{\partial L}{\partial v_i} = \alpha_i \frac{\partial L}{\partial a} + \beta_i \frac{\partial L}{\partial b},$$

$$\Delta a = \sum_i \alpha_i (v_i - \eta \frac{\partial L}{\partial v_i}) - a = -\eta ||\alpha||^2 \frac{\partial L}{\partial a} - \eta \alpha^T \beta \frac{\partial L}{\partial b},$$

$$\Delta b = \sum_i \beta_i (v_i - \eta \frac{\partial L}{\partial v_i}) - b = -\eta ||\beta||^2 \frac{\partial L}{\partial b} - \eta \alpha^T \beta \frac{\partial L}{\partial a}.$$

We can see that each concept is *not* updated in the direction of its negative gradient; rather, the updates are mixed with gradients from other concepts, where the level of entanglements is decided by $\alpha^T \beta$, i.e., how strong the hidden activations of $a$ and $b$ overlap (red neurons in Figure 1(b), Right).[5] This overlap is in turn determined by the surface form names of $a, b$ and the configuration of lower-level weights, which are also subject to change and could add further complications. Note that while our analysis here focuses on one specific layer, the effect of entanglements naturally scales with model depth as the representational distortions accumulate throughout layers. This is very problematic, especially given that concept names could be almost arbitrary and exhibit all kinds of correlations. For example, imagine two different people with somewhat similar names. While there are pattern overlaps during recognition, after it is complete, they should become two distinct objects, and the information that we wish to store on each should be stored independently and not interfere. However, as we can see, for transformers with gradient-based optimization, the overlaps in activation patterns effectively cause the learning of different concepts to "mix", which could adversely influence the training dynamics and generalization.

We note that the entanglements here are problematic only *during learning*. It is entirely fine and often beneficial for different concept representations to share latent structures, which can lead to more efficient storage and retrieval. However, it is undesirable for these shared structures, which inherit the arbitrariness of surface-form names and other correlations, to disrupt the learning itself.

## 3.3 EXPERIMENTS

We perform a series of experiments to ground the above analysis. Scientific-wise, the experiments support the previous conjectures and arguments. Practical-wise, the explorations lead to a model design based on Joint-Embedding Predictive Architectures (JEPA) (LeCun, 2022) which breaks the Reversal Curse with high performance given prior knowledge of the location of concept representations. This also unlocks a new kind of parametric memory integration that could solve large-scale arithmetic reasoning problems better than LLMs based on non-parametric memory, which we discuss in §4.

---

[4] Here we ignore the residual connection and bias terms for simplicity.

[5] These entanglements clearly extend to momentum-based updates in modern optimizers.

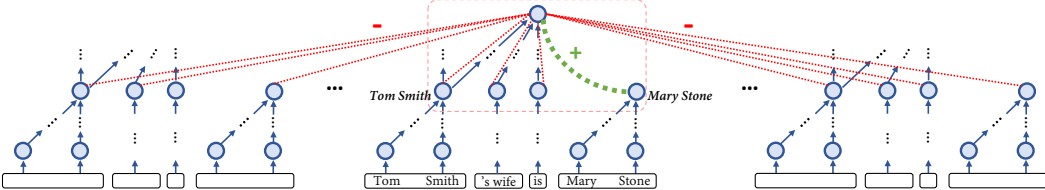

Figure 2: Illustration of JEPA with in-batch contrastive learning.

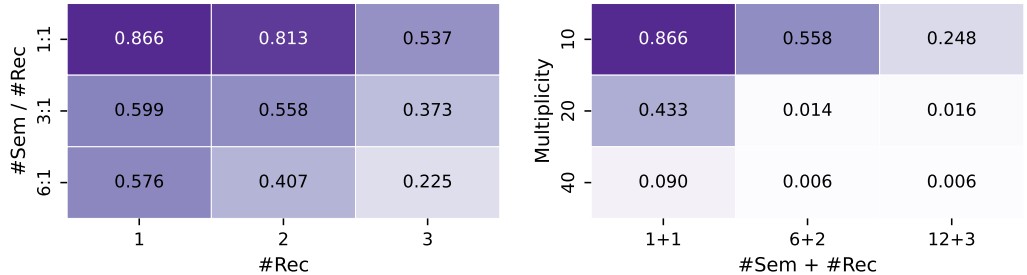

Figure 3: Performance for JEPA with in-batch contrastive loss. **Left**: performance across varying depths of the recognition module (**#Rec**) and semantic module (**#Sem**). JEPA unlocks highly non-trivial generalization, but suffers from entanglements whose effects scale with model depth. **Right**: impact of multiplicity across different model configurations. Performance consistently and significantly degrades as multiplicity increases.

**Attaching surface-level names to concepts**. We build upon the setup in the abstract setting (§2) with $|\mathcal{E}_A| = 50K$ and attach a unique surface-form name for each concept, where the inputs now become regular token sequences concatenating the concept names. Our preliminary experiments show that the names of relations do not affect the result. For the entities, we choose not to use real-world entity names since they typically do not emit meaningful statistics to experiment with. Instead, we use a simple controllable way to create overlapping names inspired by human names. Specifically, each entity name has two tokens (resembling the first name and last name of a person) belonging separately to two disjoint sets, where each entity is randomly assigned a unique (ordered) pair of tokens. We define *multiplicity* to be the number of entities who share the same first/last token, which controls the overall degree of surface name overlaps. We keep the same multiplicity for each unique token. While there are distances from realistic settings, we believe that the notion of multiplicity here is a good abstraction for the overlaps in real-world entity names. By default, we experiment with a multiplicity of 10, and also examine how varying the multiplicity affects the model's learning.

We first conduct a series of experiments on models trained with standard language modeling objectives at the surface level, and confirm that **transformers fail entirely (0.0% accuracy) to learn reversal from surface-level predictions**, regardless of architectural variants (e.g., depth, tied vs. untied embeddings). These findings align with prior work (Golovneva et al., 2024; Allen-Zhu, 2024).

Next, we study the impact of inconsistency and entanglements by deliberately scaffolding targeted modifications into the model architecture.

We begin with explicitly encouraging conceptual consistency in reversal. One key observation is that the well-known Joint-Embedding Predictive Architecture (JEPA) (LeCun, 2022), which conducts predictions at the abstract representation space instead of raw input space, could perfectly serve this purpose *if the abstract representations for prediction are at the concept level*. This is somewhat a "coincidence" since the original motivation of JEPA is to ignore unpredictable/unimportant information in the inputs, which is not related to the focus here. We experiment with a simple instantiation of JEPA based on in-batch contrastive learning, illustrated in Figure 2. Here, autoregressive prediction is done at the concept level encoded by the recognition module, where the representations of other "activated" concepts within the same batch (besides the ground truth) serve as negatives for the standard InfoNCE contrastive loss (van den Oord et al., 2019) (additional details in Appendix B). We evaluate the quality

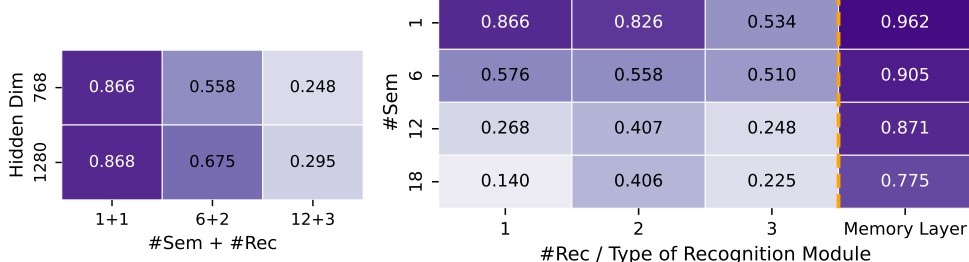

Figure 4: Mitigating the effect of entanglements by increasing the model width (**left**) and using special memory layers for the recognition module (**right**). It can be seen that increasing the model width only brings incremental improvements, while memory layers,which eliminate entanglements by design, could boost generalization by a large margin.

of learned representations by comparing the predicted state against representations of all concepts. We experiment with different configurations of the model in terms of the depth of the recognition module (**#Rec**) and the depth of the semantic module (**#Sem**), the semantic processing component on top of the recognition module.

**JEPA unlocks generalization, but suffers from entanglements.** Results are in Figure 3 (Left). *By simply encouraging conceptual consistency (with JEPA), the model can achieve highly non-trivial generalization.* To our knowledge, this is the first-ever model design that breaks the reversal curse without side-stepping the core problem. Another important observation is that the performance decreases when the model becomes deeper. Specifically, generalization consistently worsens when increasing either 1) the depth ratio between the semantic and recognition module, or 2) the overall depth of the model while keeping the ratio fixed. This strongly corroborates our analysis that the *effect of entanglements scales with model depth.* We also examine the impact of multiplicity, where the results are in Figure 3 (Right). It could be seen that *increasing the multiplicity severely hurts generalization*, especially with deeper models whose performance could drop to near zero with a mere multiplicity of 20. Overall, *the results here suggest that current models likely learn a low degree of conceptual consistency, and even if not, it is still challenging for them to learn reversal due to the effects of entanglements.*

**Mitigating entanglements.** We next explore how mitigating entanglements affects generalization, based on the JEPA design above. A straightforward strategy is to increase the *width* of the model. Intuitively, with larger hidden dimensions and more hidden units, there should be a greater chance for different concept representations to be more separated from each other. To investigate this, we train models with hidden dimensions increased from 768 to 1280, and 20 attention heads. Another approach is to build specialized recognition modules with more discriminative hidden activations. Memory layers (Sukhbaatar et al., 2015; Berges et al., 2024) exactly exemplify this approach, featuring ultra-wide hidden layers with top-$k$ sparsity and softmax activations. In particular, if we use a memory layer with small $k$ and/or high softmax temperature to replace the last MLP layer, the recognition module effectively reduces to having separate learnable embeddings for concepts with distinct names (same as the abstract setting (§2)), eliminating entanglements.[6] We experiment with this setup to see its effect on model performance.

It can be observed that increasing the width does aid generalization, but only incrementally (Figure 4, Left). Meanwhile, the specially designed memory layer could significantly enhance performance, though generalization still mildly declines with more semantic layers (Figure 4, Right). These results also confirm that the limited generalization observed with standard transformer layers as the recognition module is not due to insufficient capacity, since the module with two 1280-width transformer layers already has nearly the same amount of effective parameters as the memory layer (61.1M vs. 61.4M). We further validate that standard transformers equipped with memory layers also fail to generalize, confirming that the inconsistency inherent under surface-level prediction remains a primary blocker; memory layers are effective only when this inconsistency is resolved. Overall, these results *underscore the importance of thoughtful designs in addressing the issue of*

---

[6]Note that with the memory layer here, the depth of the recognition module does not matter.

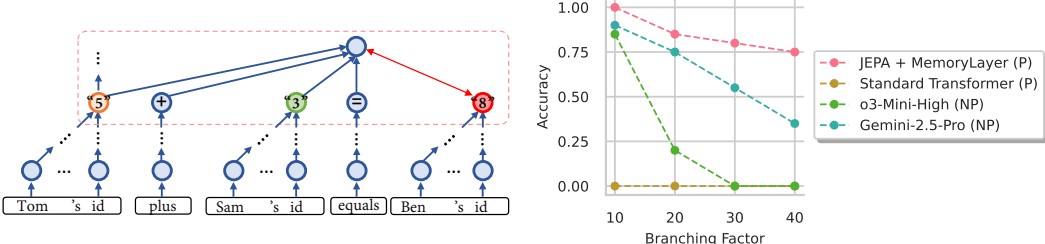

Figure 5: **Left**: illustration of the parametric variable binding enabled by models with reversal skills. **Right**: performance on the large-scale arithmetic reasoning task with various branching factors. "(P)": Parametric Memory. "(NP)": Non-Parametric Memory.

*entanglements apart from scaling.* Our findings also provide a concrete example that *memory layers can improve generalization*, corroborating recent efforts on scaling memory layers that report enhanced performance on general-domain tasks (Berges et al., 2024).

# 4 ARITHMETIC REASONING VIA PARAMETRIC FORWARD-CHAINING

Our high-level goal is to improve the *parametric memory* of current AI models, which we believe is important for handling difficult knowledge and reasoning tasks. While our previous explorations expose obstacles and pathways for models to break the Reversal Curse (and beyond), the benefits towards tackling more ambitious challenges seem rather unclear—take reversal as an example, a natural question would be: *What exactly can be achieved if the model does acquire reversal, other than knowing some more simple facts that we could have just retrieved from somewhere else?*

In this section, we show that reversal enables a new kind of parametric memory integration that allows models to solve large-scale arithmetic reasoning problems with much better performance than frontier LLMs based on non-parametric memory.

We are inspired by recent work that formalizes and scales the complexity of arithmetic reasoning problems in similar styles with popular benchmarks such as GSM8K (Ye et al., 2024; Zhou et al., 2025). An important observation is that reversal is a key skill needed for a kind of *parametric variable binding* that allows the model to infer and implicitly chain different pieces of information in parametric memory. To illustrate, imagine we are given three pieces of information: *"X equals 5"*, *"Y equals 3"*, and *"X plus Y equals Z"*. Here, $X, Y, Z$ could be any phrase that corresponds to a numerical value (prevalent in arithmetic problems), such as *"Tom's id"* or *"the amount of apples Bruce has"*. Given these facts and basic arithmetic knowledge, we could naturally know *"Z equals 8"*. Importantly, *this simple skill requires reversal to perform if we wish to store this information parametrically*, since after retrieving and adding the values of $X$ and $Y$ ($5 + 3 = 8$), a reversal step is needed to go from *"8 equals Z"* to *"Z equals 8"* (Figure 5, Left). Here, the recognition module effectively acts as a *variable-binding* module, which maps a variable name to its value.

The significance of this skill lies in its ability to not only infer unknown values, but also *propagate these inferences through a chain of deductions*: when the inferred value is properly bound to the variable, it can then serve as a stepping stone for uncovering additional unknowns that the variable connects with, triggering a cascading effect. This enables the model to perform *parametric forward-chaining* while internalizing the information, allowing it to bridge increasingly distant knowledge gaps over multiple steps in parametric memory.

**Synthesizing complex arithmetic reasoning problems.** We first conduct experiments in similar styles as in earlier sections, where we verify that the same design with JEPA and memory layers could achieve high performance on basic single-step deductions, whereas standard transformers fail completely. We then synthesize large-scale arithmetic reasoning problems to test the model's reasoning inspired by Zhou et al. (2025). Specifically, we create search trees where nodes represent variables and edges connect variables via addition. The target (unknown) variable is 3 hops away from variables with known values, and we control the problem complexity via a custom branching factor for the search tree (more details are included in Appendix C). We vary the branching factor

among 10, 20, 30, 40, corresponding to $0.4K$, $1.6K$, $3.7K$, $6.5K$ facts on average for each problem instance. We also test frontier LLMs including o3-Mini (high reasoning effort) and Gemini-2.5-Pro based on non-parametric memory and prolonged explicit reasoning, where the facts are randomly concatenated and put in context.

**Results.** As shown in Figure 5 (Right), with JEPA and memory layers, the model could achieve impressive performance higher than LLMs based on non-parametric memory. In particular, when the problem size scales, the performance drop is mild with parametric memory, while LLMs with non-parametric memory suffer more significantly. On the other hand, as expected, standard transformers consistently fail. Overall, the results here showcase the potential of well-designed parametric memory for complex reasoning problems.

## 5 RELATED WORK

**The Reversal Curse** is coined by Berglund et al. (2024), which discovers that state-of-the-art (SoTA) LLMs fail at forming reversible factual associations under both direct testing and fine-tuning settings. Similar observations are also made in Grosse et al. (2023); Allen-Zhu & Li (2025). Ma et al. (2024) finds that LLMs cannot update their knowledge in the reverse direction of knowledge editing, reinforcing this limitation. Several studies attempt to mitigate this issue through non-causal training objectives or data augmentation strategies like reversing or permuting sentence segments (Lv et al., 2024; Kitouni et al., 2024; Golovneva et al., 2024; Guo et al., 2024; Lu et al., 2024), however, these approaches side-step the fundamental problem since the two directions are still not stored as one integrated piece. Lin et al. (2024) shows that the issue may be related to inherent biases in LLMs' factual recall. Zhu et al. (2024) theoretically proves that transformers cannot learn reversal under specific settings and assumptions. Our work examines the Reversal Curse at a basic level, and to our knowledge, presents the first architectural design that truly overcomes this limitation.

**The binding problem** is a long-standing challenge in cognitive science, neuroscience, and AI. The cognitive science and neuroscience research focuses on explaining how the human brain solves this problem (Roskies, 1999; Engel & Singer, 2001; Zimmer et al., 2006), while AI studies investigate how to achieve adequate binding in artificial neural networks (Greff et al., 2020). There are two major types of binding: perceptual binding and conceptual binding; related literature is discussed in §3. Extensive research demonstrates that transformers effectively learn perceptual binding (Meng et al., 2022; Geva et al., 2023; Lad et al., 2024; Yang et al., 2024; Feng & Steinhardt, 2024; Kaplan et al., 2025; Pal et al., 2023; Belrose et al., 2023; Wu et al., 2024; Cai et al., 2024). In this work, we identify two major limitations in transformers' conceptual binding that potentially cause the Reversal Curse, and demonstrate that explicitly addressing them through targeted designs enables models to break the Reversal Curse with high performance.

## 6 DISCUSSION & CONCLUSION

We conjecture that the Reversal Curse in LLMs is caused by *inconsistency* and *entanglements* of concept representations, two aspects of the long-standing *binding problem* in cognitive science, neuroscience and AI. A series of experiments supports our hypotheses, and leads to model designs that could break the Reversal Curse with high performance. It is important to note, however, that these fundamental issues underlying the Reversal Curse that we identify are far from being resolved, since *our current solutions rely heavily on human scaffolding and are specifically tailored to the reversal task*, which only deals with the most basic concepts. For instance, we need prior knowledge of the concept locations for the JEPA design to promote consistency, and the design only fosters consistency in a highly restricted manner (where concepts emerge as perceived subjects and predicted objects). Similarly, the memory layer design leverages our prior knowledge that each unique name indeed corresponds to a unique concept in our setting, and it would impede learning in cases where synonymy exists. Overall, the more fundamental challenge lies in designing *systematic conceptual binding mechanisms with less human scaffolding*, applicable to more abstract concepts and complex skills (Sutton, 2019). Our explorations rigorously establish these challenges to the broader community. Finally, we demonstrate that reversal skills unlock parametric forward-chaining, enabling models to solve large-scale arithmetic reasoning tasks better than frontier LLMs using non-parametric memory.

## ETHICS STATEMENT

The research presented in this paper adheres to the ICLR Code of Ethics. Our work is foundational, focusing on a specific limitation of transformer models known as the Reversal Curse. We do not use any datasets that contain private, sensitive, or personally identifiable information. The study does not involve human subjects, and we do not foresee any direct negative societal impacts or ethical concerns arising from our methodology or findings.

## REPRODUCIBILITY STATEMENT

To ensure the reproducibility of our findings, we have provided detailed descriptions of our experimental setup throughout the paper. The source code used for all experiments will be made publicly available upon publication.

- **Data Generation**: The procedure for generating the training and evaluation datasets for the reversal task is detailed in §2 and §3.3. The method for synthesizing the large-scale arithmetic reasoning problems is described in §4, with further details in Appendix C.
- **Model Architecture**: We describe the architectures of the models used, including standard transformers, JEPA-based models, and models incorporating memory layers, in §2 and §3.
- **Training Details**: training details and hyperparameters for our experiments are provided in Appendix A.

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

## A    HYPERPARAMETERS AND TRAINING DETAILS

We use the standard transformer architecture as in GPT-2 (Radford et al., 2019), with 768 hidden dimension, 12 attention heads and no positional encoding unless otherwise specified. For optimization, we use the AdamW optimizer (Loshchilov & Hutter, 2019) with 2000 warm-up steps, learning rate $1e-4$, weight decay 0.25. For experiments on reversal curse (§2, §3), we use batch size 512 or 1024 and evaluate the models every 50000 optimization steps. For experiments on arithmetic reasoning (§4), we use batch size 128 and evaluate the models every 20000 optimization steps. All implementations are based on PyTorch (Paszke et al., 2019) and Huggingface Transformers (Wolf et al., 2020). Model trainings are done on NVIDIA A6000 and A100 GPUs.

## B    ADDITIONAL DISCUSSION & RESULTS ON JEPA

**JEPA Framework.**    The Joint-Embedding Predictive Architecture (JEPA) was proposed by LeCun (2022) as a departure from conventional generative architectures. Its core premise is that prediction should be performed in the abstract representation space of *encoded* inputs rather than in the raw input space. This is motivated by the observation that precise reconstruction of raw inputs is often unnecessary or impossible, and that effective learning typically focuses on structures within high-level abstractions. While JEPA has been predominantly applied to vision domains such as images and videos (Assran et al., 2023; Bardes et al., 2024), recent work has begun exploring similar methodologies in language domains (Barrault et al., 2024). In the context of our study, we draw a direct parallel between the functional stratification of LLM layers and the modular components of the JEPA framework:

- **Recognition Module as the Encoder:** As discussed in the main text, numerous prior works demonstrate that the lower layers of LLMs perform "detokenization"—mapping surface-level subword tokens into cohesive concept representations. Within the JEPA framework, this aligns with the *Encoder*, which is responsible for mapping raw inputs into latent abstract representations.

- **Semantic Module as the Predictor:** The middle and upper layers of the LLM operate on this concept space to perform abstract reasoning. In the JEPA framework, this corresponds to the *Predictor*, which operates within the latent space to predict target concept representations without reverting to the raw input space.

**Experimental Stability.** We conducted a stability analysis by repeating the experiments corresponding to Figure 3 (Left) using two additional random seeds (for a total of three runs) to assess the variance of the results. Table 2 reports the mean and standard deviation for varying configurations of the depths of the Recognition and Semantic modules. The results exhibit low standard deviation across all settings, confirming that the trends reported in our study are stable and robust to initialization.

Table 2: Stability results of JEPA experiments (corresponding to Figure 3, Left). We report the mean ± standard deviation across 3 random seeds.

|  | #Rec = 1 | #Rec = 2 | #Rec = 3 |
|---|---|---|---|
| **#Sem/#Rec = 1** | $0.864 \pm 0.008$ | $0.814 \pm 0.003$ | $0.533 \pm 0.004$ |
| **#Sem/#Rec = 3** | $0.594 \pm 0.006$ | $0.565 \pm 0.006$ | $0.371 \pm 0.009$ |
| **#Sem/#Rec = 6** | $0.573 \pm 0.003$ | $0.402 \pm 0.012$ | $0.230 \pm 0.004$ |

## C   ARITHMETIC REASONING

### C.1   LEARNING BASIC ARITHMETIC DEDUCTIONS

We first test whether models can learn to perform single-step arithmetic deductions as described in §4. To reiterate here, for variables $X, Y, Z$ (whose "names" are now phrases which correspond to numerical values), we hope that the model could learn *"Z equals* 8*"* after internalizing *"X equals* 5*"*, *"Y equals* 3*"* and *"X plus Y equals Z"* (apparently, also for other numerical values). Note that the key challenge here is not the numerical calculations, rather, it is the *variable binding* which requires reversal to perform.

We follow the setup in §3, with changes on the train/test data. Since our focus (and also the focus of arithmetic reasoning problems in general) is not on numerical calculations, we add the constraint that for all additions, at least one of the two left-hand-side arguments must be $m$ or $n$, which are two small distinct predetermined integers randomly chosen from $[10, 50]$. In other words, all calculations only involve adding $m$ or $n$ (instead of all possible values) to some value. We also operate under modular arithmetics with $P = 10007$ to avoid under/overflows. To synthesize the training data for learning the rules, for each possible value $i$ in $[0, P-1]$, we prepare 10 distinct variables that are assigned value $i$ (i.e., adding into the training set the fact *"X equals i"* for each variable $X$). Then, we add a random 30% of all relational facts that satisfy the previous constraint that could be formed among the variables. For testing, we first randomly select some variable pairs where at least one variable in each pair has value $m$ or $n$ (*s.t.* the calculation is "taught" during training). Then for each variable pair $(X, Y)$, we create a *new* variable $Z$, add *"X plus Y equals Z"* in the training set and *"Z equals \_\_"* in the test set, to evaluate whether the model can infer and store the correct value of $Z$. Variable names are generated the same way as in §3 with two tokens each and multiplicity 10. This name assignment is also conceptually similar to the ones in GSM-$\infty$ (Zhou et al., 2025) such as entity attributes (e.g., *"number of tigers in Hamilton Farm"*).

We find that the same design which breaks the Reversal Curse with JEPA and memory layers (§3) could generalize decently, achieving an MRR of 0.718 with 6 semantic layers. On the other hand, expectedly, models that predict at the surface level fail to generalize.

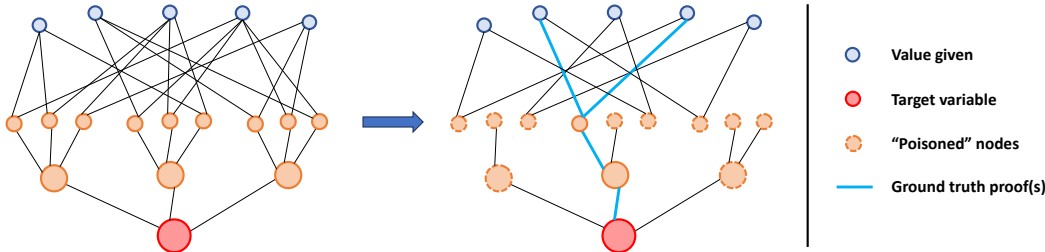

Figure 6: Illustration of the problem synthesis for large-scale arithmetic reasoning. For simplicity, we omit the nodes with given values used for connecting variables via addition.

## C.2 SCALING THE REASONING CHALLENGE

We use a simple way to synthesize problems with different scales, by first generating a complete tree, and then dropping a portion of the edges to form a search problem (Figure 6).[7] Specifically, the complete tree has a fixed depth 3, where 1) each node represents a variable, where the root node (at the first layer) is the target with a randomly chosen integer value to be inferred by the model; 2) each edge connects two variables via addition through another variable with a given value $m$ or $n$ (randomly assigned). The first two layer nodes have a custom branching factor $(10, 20, 30, 40)$ and the third layer nodes have a fixed branching factor of 6 connecting to leaf nodes with given values (decided by the variable values along the paths). For each problem instance, we randomly choose the value of the target node from $[200, 800]$, which ensures that all numerical calculations involve small positive integers (below $1000$) that LLMs can perfectly perform. Note that with the complete tree, the target value could be inferred by following any path from any leaf node. We drop a portion of the edges to create a reasoning challenge. Concretely, we "poison" $60\%$ of second and third layer nodes by breaking paths through them between the target and leaf nodes: for each third layer node which itself or its parent is poisoned, we randomly drop either the edge connecting it to its parent, or all edges connecting it to the leaf nodes. Deriving the target value is, in essence, a "path-finding" problem where the model needs to find paths connecting the target with any of the leaf nodes (whose values are given). This is simpler than the more general "graph-finding" problem in arithmetic reasoning problems (Zhou et al., 2025), but still challenging when the search space grows large.

We synthesize 20 instances for each branching factor for testing. For models with parametric memory, since we train the models from scratch, we merge the problem facts with the training data in §C.1 with disjoint variable names to teach the model basic arithmetics and deductions. For testing LLMs with non-parametric memory, we use a very simple template and match each variable with a distinct human name, e.g., *"Tom's number is* 5*"* and *"Tom's number plus Amy's number equals Bob's number"*, with no commonsense or other implicit knowledge involved. The specific choice of the template marginally affects LLM performance from preliminary tests.

**Errors of LLMs.** We examine error cases of LLMs to understand their failure modes. For o3-Mini-High (which does not return the thinking tokens), the model summarizes the thinking processing at a high level with statements such as *"Every acceptable solution of the many equations forces..."* and *"One may check by solving the huge simultaneous network of sum-equations that..."*, and hence it is difficult to pinpoint the specific errors. For Gemini-2.5-Pro, we find that the model never makes calculation errors, and among 10 random error examples, 7 stem from making (wrong) guesses without thorough consistency checks, 2 are caused by hallucinating unprovided facts, and 1 from a copy error. Overall, LLMs seem to struggle with forming integrated/compressed representations of information provided in context, and have to rely on extensive explicit search to recognize the connections between different pieces of information. With well-designed parametric memory, on the other hand, the facts could be more tightly connected and integrated, which enables models to solve the challenge with better performance and milder performance drop as problem scales increase.

---

[7]Technically, a "tree" is not an accurate description of the network since the "leaf" nodes here could have multiple parents; we abuse the term for simplicity.

## D    USE OF LARGE LANGUAGE MODELS

Large Language Models (LLMs) were used solely as a general-purpose writing aid in the preparation of this paper. Specifically, they were used to help polish grammar and improve the clarity of certain sentences. No LLMs were used for research ideation, experimental design, data analysis, or drawing conclusions. All substantive contributions to the research and writing were made by the authors.

