# OpenReview forum: "Is the Reversal Curse a Binding Problem? Uncovering Limitations of Transformers from a Basic Generalization Failure"
_ICLR.cc/2026/Conference — ICLR 2026 Poster_

### Official Review · Reviewer_NLXN · 2025-10-24

**Soundness:** 3
**Presentation:** 2
**Contribution:** 2
**Rating:** 4
**Confidence:** 3

**Summary:**

The paper investigates the "Curse of Reversal" problem in LLM that was pointed out in a previous study, where a model that learned "A is B" cannot generalize it to "B is A".  The paper argues that this problem boils down to the binding problem in the sense that it arises from inconsistency representations of the same concept and entangled representations of separate concepts.  The paper demonstrates this experimentally by introducing JEPA with contrastive learning and showing that this can overcome the reversal problem in their smallish setting; they additionally applies the method to arithmetic tasks.

**Strengths:**

The paper addresses interesting question about deficiency of reversal in LLM.  Inconsistency and entanglement are not novel by themselves but common worries about representation in neural networks.  However, connection with reversal problem is an interesting and novel observation.  They provide  demonstrating experiments that are convincing in their smallish settings.

**Weaknesses:**

As mentioned, inconsistency and entanglement are common worries about representation in neural networks.  Usually, one expects that big data can overcome this, but somehow reversal problem still exists in LLM.  My main concern is that, although they demonstrated the connection in toy tasks, it's not clear how much valid their argument is with respect to LLM (line 353-355).  They do not seem to materialize on this point, and this is rather disappointing since they emphasize this in Introduction.

Clarity issue:  The authors should briefly introduce JEPA in their context as their experiment relies on this.   From what's written, it's not clear what are recognition and semantic modules in Transformer architecture and Figures 3, 4 are hard to understand.  (I looked at JEPA paper, but still didn't understand precisely because the presentations are not aligned.)

**Questions:**

In page 7, the authors present the result that, when the model is larger, the model suffers from entanglement even with JEPA.  What's the significance of this result?  Isn't it just usual overfitting?  Since the dataset is small, this result seems to be well expected.  Is this what happens in LLMs, where the dataset is huge?

---

> ### Author Response · Authors · 2025-11-21
>
> We thank the reviewer for the constructive feedback and for recognizing the novelty of connecting the Reversal Curse to the classical binding problem. We address the concerns/questions below.
>
> **1. Validity of Argument for LLMs**
>
> The reviewer raises an important point: if large datasets typically overcome representation issues, why does the Reversal Curse persist in LLMs trained on trillions of tokens? This persistence suggests the failure stems not from insufficient data, but from fundamental structural limitations in how standard Transformers store and retrieve relational knowledge.
> To understand this problem rigorously, we must isolate its underlying causes. Studying it directly on LLMs trained on uncontrolled corpora makes it nearly impossible to disentangle architectural limitations from data distribution issues. Therefore, we adopt a controlled scientific approach (akin to the Physics of Language Models series):
> - **Hypothesis**: Through analysis, we hypothesize that the failure stems from the lack of appropriate binding mechanisms (inconsistency and entanglement) under surface-level predictions.
> - **Controlled Experiment**: We design controlled settings that isolate these variables.
> - **Validation**: We show that addressing these specific variables (via JEPA and Memory Layers) breaks the curse where standard Transformers fail.
>
> While our experiments in controlled settings cannot definitively prove that the same mechanism operates identically in LLMs, they provide: (1) a rigorous explanation for why the curse persists despite massive data, and (2) a working solution that effectively breaks the curse, with demonstrated applications to complex arithmetic reasoning problems. We believe this is significant initial progress towards better understanding this problem and its impact in more realistic settings.
>
> **2. Clarity on JEPA, Recognition, and Semantic Modules**
>
> We apologize for the lack of clarity. To clarify:
> - **Recognition Module (Lower Layers)**: As discussed in Lines 179–185, numerous prior works demonstrate that lower layers of LLMs perform "detokenization"—mapping surface-level subword tokens into cohesive concept representations, effectively “recognizing” the concepts. In the JEPA framework, this corresponds to the **Encoder**, which maps raw inputs to latent abstract representations.
> - **Semantic Module (Middle and Upper Layers)**: These layers operate on the concept space to perform abstract “thinking” and reasoning. In the JEPA framework, this corresponds to the **Predictor**, which operates in latent space to predict target concept representations.
> - **JEPA Architecture**: We adopt the standard JEPA architecture with in-batch contrastive loss. Here, autoregressive prediction is done at the concept level encoded by the recognition module, where the representations of other “activated” concepts within the same batch (besides the ground truth) serve as negatives for the standard InfoNCE contrastive loss. We also provided an illustration in Figure 2.
>
> We will expand these explanations in the revision to ensure these concepts are clearer.
>
> **3. Entanglement vs. Overfitting**
> - The reviewer asks whether entanglement in larger models is simply "usual overfitting." We argue it is mechanistically distinct:
> In our abstract setting (Section 2), standard Transformers generalize strongly even with deep models (18 layers) trained on very small datasets (2.5K entities), with **no generalization degradation as depth increases**. However, in surface-name settings with a large dataset (50K entities), deeper models show clear performance drops.
> - This pattern is inconsistent with standard overfitting, which would affect both settings similarly when model capacity becomes larger. Instead, it aligns precisely with our entanglement analysis: distortions of concept representations caused by entanglement accumulate through layers, and hence further hurt generalization as model depth increases. While this failure could still be described as "overfitting," our work pinpoints *why* it occurs and demonstrates the architectural designs that can mitigate it.
>
> We thank the reviewer again for the thoughtful comments and constructive suggestions. We will incorporate the improvements and additional results into the revised draft over the next few days. We look forward to addressing any additional questions or comments.

---

> > ### Comment · Reviewer_NLXN · 2025-11-27
> > **Reply to authors**
> >
> > Thanks to the authors.  The clarity issues seem to have been addressed.
> >
> > However, I still have concerns about how the authors' hypothesis is valid in the large scale setting, although they provided controlled experiments in the small scale setting.  There could be some phase transition, while increasing the scale, where the model might change the representational strategy, so that the authors' argument seems to me not convincing enough.   I keep my score.

---

> > > ### Author Response · Authors · 2025-11-27
> > > **Reply to Reviewer NLXN**
> > >
> > > Thank you for the follow-up. We are happy to hear that the clarity issues have been resolved.
> > >
> > > Regarding phase transitions: while such phenomena can indeed occur in deep learning, the Reversal Curse has been documented extensively in large-scale LLMs across multiple works (Berglund et al., 2024; Grosse et al., 2023; Allen-Zhu & Li, 2025; Ma et al., 2024). These studies consistently show that scaling (even to hundreds of billions of parameters) does not induce a representational shift that resolves the issue. In our controlled setting, we also swept over model sizes (up to 18 layers) and dataset sizes (up to 100K entities), and observed smooth trends with no evidence of divergence or emergent representational changes (except those expected such as the issue of entanglement). Given this, we would appreciate clarification on whether you still see a concrete concern about potential representational changes at larger scales. If so, what scale would you consider informative (e.g., 7B/13B)?
> > >
> > > We also note that our architecture is, to our knowledge, the first to break the reversal curse, and that the reversal skill already finds applications in domains with *formal* structures (e.g., arithmetic reasoning; Section 4) without needing to be integrated into existing large-scale language models. In terms of integration with LLMs, we have also discussed potential strategies and challenges in our response to Reviewer 4J4R under “Generalization to Realistic NLP Settings.” We believe this is a very important direction, however, it is beyond the scope of the present work, which focuses on a clean and in-depth investigation of the reversal curse itself.

---

### Official Review · Reviewer_3p1K · 2025-10-31

**Soundness:** 3
**Presentation:** 3
**Contribution:** 3
**Rating:** 6
**Confidence:** 3

**Summary:**

This paper addresses the Reversal Curse, a well-known challenge for large language models (LLMs). The authors argue that this limitation stems from an issue related to concept-level binding, rather than being solely due to insufficient data or architectural constraints. To examine this hypothesis, the authors design two controlled toy tasks—one that uses concept-level representations for inverse relations and another that attaches surface-level names to those concepts. Through these experiments, they identify two potential causes of the models’ failure to generalize inverse relations: (a) inconsistency in how concepts are represented across contexts and (b) entanglement between the representations of different concepts. To address these issues, the paper introduces a model based on JEPA-style joint embedding learning and in-batch contrastive learning, enhanced with memory layers to reduce representational interference. The proposed model successfully resolves the Reversal Curse in experiments and further shows that strengthening conceptual binding is crucial for inverse relational generalization.

**Strengths:**

- **S1.** By designing and evaluating two different types of controlled toy tasks, the paper effectively demonstrates limitations of current LLM architectures and clearly motivates why the Reversal Curse is closely tied to conceptual binding.
- **S2.** The paper relates classical cognitive science concepts—such as the binding problem—to modern transformer shortcomings, offering valuable insights to the community.
- **S3.** The authors propose a method that is well-aligned with the identified problem and show clear improvement in overcoming the Reversal Curse through experiments.
- **S4.** The paper extends beyond simple toy tasks and demonstrates the usefulness of resolving inverse relational generalization in more complex settings, such as arithmetic reasoning, illustrating the practical benefits for LLMs.

**Weaknesses:**

- **W1.** Although the paper attempts to show effectiveness beyond toy tasks by including an arithmetic reasoning task, it remains unclear whether the results generalize to more realistic real-world settings.
- **W2.** The JEPA approach introduced to address inconsistency requires prior knowledge of concepts, yet defining such concepts in practical domains is challenging. This could make it difficult to scale the method to real-world applications.
- **W3.** While the experiments investigate the effect of entanglement using the multiplicity setting, there is no analysis of whether the proposed memory layers remain effective under high levels of multiplicity. Including such results would help clarify the role of memory layers in reducing representational entanglement.
- **W4.** The paper lacks sufficient ablation studies on the configuration of the JEPA module and memory layers. More comprehensive ablations enable readers better to understand the contribution and importance of each component.

**Questions:**

- **Q1.** Using JEPA to address the conceptual binding problem appears promising. However, in real-world applications where prior conceptual knowledge is limited, how do the authors envision applying or extending this approach? I would appreciate the authors’ perspective on practical strategies for such scenarios.
- **Q2.** In line 348, the paper claims: “This strongly indicates that the effect of entanglements scales with model depth.” Could the authors clarify how the results in Figure 3 specifically support the claim that the observed issue is driven by entanglement?
- **Q3.** Beyond the need for prior knowledge, what computational or optimization challenges might arise when integrating the proposed methods into existing large-scale transformer architectures?

---

> ### Author Response · Authors · 2025-11-21
>
> We thank the reviewer for the detailed and constructive evaluation. We are encouraged that you find value in our connection between the classical binding problem and the Reversal Curse (S2), the alignment of our JEPA-based solution with the diagnosed causes (S3), and our demonstration of reversal skills in complex arithmetic reasoning (S4). We address your concerns below.
>
> **W1: Generalization to realistic settings**
>
> Indeed our primary experiments adopt simple and controlled settings. This choice is intentional: isolating the mechanisms behind the Reversal Curse requires experimental control that is near impossible with models trained on uncontrolled corpora. That said, our results already take a meaningful step beyond toy tasks - as shown in Section 4, the ability to learn reversible associations enables our model to solve complex arithmetic reasoning problems with competitive performance stronger than frontier LLMs. Note that this is not merely a math task; it acts as a proxy for *variable binding and tracking*, a critical capability for many realistic logic-heavy tasks. Our findings suggest promising future directions on extending the applications to complex problems with more diverse primitives.
>
> On the other hand, we note that recent and concurrent work has successfully applied JEPA-style architectures and memory layers to large-scale language modeling, showing promising results (e.g., Huang et al., 2025; Lin et al., 2025). Our work is orthogonal to these efforts, where we provide more rigorous understanding and justifications for why these architectural designs are important for resolving generalization failures like the Reversal Curse, and concrete future challenges to tackle.
>
> **W2 & Q1: Dependence on prior knowledge**
>
> As we also acknowledged in Section 6, our current solution based on JEPA relies on prior knowledge of the location of concept representations and name-concept correspondence. However, this was a design choice for experimental rigor, not an inherent limitation of the architecture. We envision two strategies for more practical deployment/integration of the designs:
> - Implicit Induction: JEPA has been proven successful in learning abstract representations *without* explicitly setting the labels/abstraction levels (e.g., I-JEPA, V-JEPA). In scaled-up uncontrolled settings, we do not need to manually define the "concepts" and where they need to live; instead, the bottleneck abstraction layer can be treated as a tunable hyperparameter, allowing the objective to induce the necessary abstractions from data.
> - If we are applying the design to some pretrained LLM, we could leverage interpretability techniques to identify the locations of concept representations. There is a long series of interpretability work (also discussed in lines 179-185) that successfully locates and manipulates concepts in language models via causal tracing/activation patching (e.g., Meng et al., 2022; Kaplan et al., 2025), which can be utilized for the JEPA design.
>
> **W3: Memory layers under high multiplicity**
>
> Mechanistically, Memory Layers are explicitly designed to remain effective under high multiplicity/overlap in input representations. By adopting sparse activations, the layer forces the model to select specific "slots" for information storage. For example, simply setting $k=1$ in the top-k activation would force the model to use a single slot for each unique concept, effectively eliminating entanglement regardless of input multiplicity. The trade-off is that highly peaked activations reduce the model's potential to learn statistical features that *should* be shared across concepts. Overall, selecting the appropriate sparsity level to balance separation and sharing is still an open problem in sparse learning, and we hope our analysis of entanglements offers insights for this ongoing research in the community.
>
> **W4: Ablation studies**
>
> We have performed substantial ablations regarding the core components of our hypothesis, specifically, we ablate the depths of the recognition and semantic modules in JEPA (Figure 3, Left), how they interact with different multiplicity factors (Figure 3, Right), as well as recognition modules with varying widths and Memory Layers (Figure 4). These ablations were conducted to isolate the effects of different components and how they are influenced by the data distribution, thereby supporting our core hypotheses. If the reviewer has specific additional ablations in mind that could be critical for validating our claims, we are happy to run them during the discussion period.
>
> (continuing in follow-up comments)

---

> > ### Author Response · Authors · 2025-11-21
> >
> > **Q2: Entanglement scaling with depth**
> >
> > Our analysis indicates that entanglements cause distortions of concept representations that accumulate throughout the layers (Section 3.2, lines 228-253). This strongly aligns with the empirical results in Figure 3 (Left) that deeper models exhibit worse generalization on the reversal task. While it is difficult to eliminate all confounding factors (e.g., deeper networks may be naturally harder to optimize), the sharp degradation strongly points to entanglement as the primary driver.
> >
> > **Q3: Challenges in integration**
> >
> > We see two main challenges when integrating the proposed designs to general-purpose LLMs:
> > - Objective Integration: In our study, the corpus consists of purely reversible facts. In more realistic settings, we must balance the "reversal" objective with the regular next-token prediction objective. Determining the optimal weighting to ensure reasoning/abstraction capabilities do not degrade linguistic fluency and other regular capabilities is a key open question; there is also recent work such as Huang et al. which shows improved performance when applying JEPA to LLM with an appropriate balance between the objectives.
> > - Training Stability: In-batch contrastive learning (used in JEPA) can be sensitive to batch size and negative sampling strategies. Stable learning would typically require robust hard-negative sampling (e.g., similar to those in the seminal DPR work by Karpukhin et al.) to prevent the model from learning trivial shortcuts.
> >
> > We thank the reviewer again for the thoughtful comments and constructive suggestions. We will incorporate the improvements into the revised draft over the next few days. We look forward to addressing any additional questions or comments.
> >
> > **References**
> > - Huang et al. LLM-JEPA: Large Language Models Meet Joint Embedding Predictive Architectures. arXiv-25.
> > - Lin et al. Continual Learning via Sparse Memory Finetuning. arXiv-25.
> > - Meng et al. Locating and Editing Factual Associations in GPT. NeurIPS-22.
> > - Kaplan et al. From Tokens to Words: On the Inner Lexicon of LLMs. ICLR-25.
> > - Karpukhin et al. Dense Passage Retrieval for Open-Domain Question Answering. EMNLP-20.

---

### Official Review · Reviewer_4J4R · 2025-10-31

**Soundness:** 3
**Presentation:** 3
**Contribution:** 3
**Rating:** 6
**Confidence:** 3

**Summary:**

This paper investigates the Reversal Curse, a fundamental generalization failure where LLMs trained on “A’s wife is B” fail to infer “B’s husband is A.” The authors argue that this limitation stems from the long-standing binding problem in cognitive science, specifically due to two factors: (1) inconsistency of concept representations when entities switch roles between perceived subjects and predicted objects, and (2) entanglement of concept representations during gradient-based learning (Section 3.2). Critically, the paper demonstrates that standard transformers can learn reversal when inputs are represented at the abstract concept level (Table 1), establishing that the problem arises specifically from surface-level predictions of autoregressive next token prediction. To address these issues, they propose a Joint-Embedding Predictive Architecture (JEPA) (LeCun, 2022), which achieves high reversal performance (Figure 3). They further show that incorporating memory layers (top-k sparsity and softmax activations) dramatically improves generalization by reducing entanglement (Figure 4), and they demonstrate that such insight enables parametric forward-chaining for solving large-scale arithmetic reasoning problems (Figure 5). Overall, the paper explored fundamental limitations of current LLM approach in the lens of binding problem.

**Strengths:**

1. Strong narrative and step-by-step improvement. The work deliberately demonstrates the current problem of the transformer-based LLM Reversal Curse, identifies that the issue comes from surface-level prediction rather than architecture itself, then introduces JEPA as a way to address representational inconsistency and finally introduces the memory layer to address entanglement. The storyline is concrete and provides a clear formulation of the Reversal Curse and current transformer-based models’ limitations. This is likely to be insightful for many LLM researchers.

2. The results are overall clearly presented. Figures and tables are clearly paired with take-away messages, and in general the empirical story is easy to follow.

3. The paper clearly state limitations in conclusion section. For example, the authors explicitly acknowledge that “our current solutions rely heavily on human scaffolding and are specifically tailored to the reversal task, which only deals with the most basic concepts,” and they frame “scaffolding, applicable to more abstract concepts and complex skills” as an open problem rather than claiming it is solved.

**Weaknesses:**

1. Motivation and uniqueness of JEPA are underspecified. The paper motivates JEPA as a fix for role-dependent inconsistency in concept representations, but it does not clearly justify why JEPA in particular is the right solution. There are many possible ways to enforce role-invariant bindings on top of current LLMs (e.g., fine-tuning, RL, etc). The paper would be stronger if it compared JEPA against alternative  appraochesor at least explained why those alternatives were ruled out.

2. Memory layer design feels somewhat ad hoc. The proposed memory layer (top-k sparse retrieval + softmax activation) is claimed to reduce entanglement and improve generalization, but it is not obvious why this particular mechanism was chosen over other forms of memory (e.g., retrieval, external KV cache persistence, etc.). The paper should clarify the choices being memory

3. Limited evidence of generalization to realistic NLP settings. The core problem (reversal) is motivated as a real failure mode of large language models trained on natural language, but most comparisons are then made to purpose-trained JEPA/memory systems on narrowly scoped reversal or arithmetic forward-chaining tasks. It is still unclear whether JEPA + parametric memory can coexist with broad natural-language behaviors like in-context learning, continual knowledge updating, or general QA. In particular, adding parametric memory might reduce flexibility (e.g., by hardening stored associations and making it harder to correct or update them online), which could degrade other capabilities.

4. There is also no report of variance across random seeds, no error bars, and no discussion of robustness. This makes it difficult to assess stability and reproducibility of Figures 3–5, and it weakens claims about systematic improvements.

**Questions:**

I am inclined to increase the score if these big-picture questions are convincingly addressed.

1. Why JEPA? There can be multiple ways to solve role-inconsistent concept representations when entities switch between subject and object positions. Why JEPA specifically? Did you consider alternative architectural or training frameworks, and if so, what failed?

2. Why the memory layer? Why this specific top-k + softmax parametric memory layer? Are there simpler or more standard memory mechanisms that would work as well? How sensitive is performance to this design?

3. Generalization to realistic settings. My biggest concern is whether JEPA + parametric memory can be scaled to standard natural-language-style data. A potential issue with a rigid parametric memory is that it could reduce flexibility on new knowledge, which might hurt other LLM capabilities. Can you provide intuition or preliminary evidence for how reducing inconsistency and entanglement will affect these broader capabilities? I would be strongly convinced by even small-scale preliminary results, or by a clear plan to test this.

Minor questions
1. What is epsilon_A in Table 1? Please define it in the caption or main text.

2. You claim that transformers “fail entirely to learn reversal” (lines 309–312). Is there quantitative evidence for “entirely”? Please provide a figure or table with those baseline numbers.

3. In Figure 5, do you also evaluate a standard transformer augmented with the same memory layer? If not, please clarify why not; this matters for isolating where the improvement comes from.

4. Do you have multiple random seeds and standard deviations / error bars for Figures 3, 4, and 5? If not, can you comment on stability across runs?

---

> ### Author Response · Authors · 2025-11-21
>
> We thank the reviewer for the thoughtful assessment and for highlighting our work’s strong narrative and clear formulation. We are encouraged that you find our step-by-step approach—from identifying the conceptual binding issues underlying surface-level predictions, to addressing them through JEPA and Memory Layers—clear and logical.
>
> **1. Why JEPA?**
>
> Our choice of JEPA is not arbitrary; it is derived directly from our diagnosis of the core problem.
> - **JEPA is a direct solution to conceptual inconsistency**: JEPA is defined by performing predictions in the abstract representation space rather than the raw input space. This offers a direct architectural solution to the representational inconsistency problem identified in our analysis.
> - **Alternatives**: We explored adding auxiliary losses (e.g., minimizing cosine distance between subject and object representations) and observed some positive initial signals. However, these are effectively variants of "joint-embedding"-based approaches, and also introduce additional hyperparameters (e.g., weighting factors) that complicate the experiments and potentially obscure the core findings. JEPA provides the cleanest framework for handling inconsistency through directly enforcing prediction in the concept space.
> - **Fine-tuning or RL**: Regarding the suggestion of fine-tuning or RL, it is not immediately clear to us how these strategies would resolve the issue of conceptual inconsistency, provided the model still relies on surface-level predictions. In our experiments, we found that training standard Transformers on massive datasets (100K entities with bidirectional facts) still yields zero generalization. Standard fine-tuning optimizes the same objective that fails to capture reversal; in fact, prior work such as the original Reversal Curse paper (Berglund et al.) also conducted fine-tuning experiments and observed that there was no generalization. For RL, due to the short-horizon nature of the problem here, the objective is similar to regular SFT. If the reviewer has specific RL or fine-tuning formulations in mind that could fundamentally alter the underlying representational structure, we would appreciate those insights.
>
> **2. Why Memory Layers?**
> - **A natural solution to conceptual entanglement**: Our theoretical analysis (Section 3.2) suggests that entanglement arises from "concept-mixing" gradients in densely-activated recognition layers. To solve this, it requires a mechanism that enforces sparsity and reduces activation overlap during parametric learning. The top-k + softmax activation design is a standard one in memory layers (see, e.g., Berges et al. & Lin et al.), which implements this requirement precisely by creating sparse activations that minimize interference, and hence we adopt it in our experiments.
> - **Parametric vs. Non-Parametric memory**: We did not consider options like retrieval or external KV caches which augment the model with external (non-parametric) memory, since our goal (and the Reversal Curse problem) is specifically about reversible associations over *parametric* memory/knowledge.
> - **Alternatives Explored**: We did explore alternative methods to handle entanglement, specifically an idea of turning concept representations into temporary “fast weights” during optimization, inspired by Difference Target Propagation (Lee et al., 2014). Theoretically, this could address representational entanglement as conceptual learning would not depend on the lower-level recognition layers (the root of entanglement from our analysis). However, the main technical bottleneck was that modern optimizers (like AdamW) rely heavily on momentum terms that are difficult to design for dynamic fast weights, leading to training instability. Memory Layers, in contrast, provided a robust, architectural solution to the same problem without these optimization challenges.
>
> (continuing in follow-up comments)

---

> > ### Author Response · Authors · 2025-11-21
> >
> > **3. Generalization to Realistic NLP Settings**
> >
> > We acknowledge that our experiments focus on controlled, rigorous settings to isolate the mechanism of the Reversal Curse. However, there is strong evidence that these design components scale to realistic NLP tasks:
> > - **JEPA in NLP**: Recent work such as Large Concept Models (Barrault et al., 2024) and LLM-JEPA (Huang et al., 2025) demonstrates that JEPA-style objectives are effective for broad language modeling and reasoning tasks.
> > - **Memory Layers in NLP**: Berges et al. (2024) show that Memory Layers can be scaled effectively in broader language domains with strong performance. Furthermore, recent work like Lin et al. (2025) demonstrates that Memory Layers allow models to update knowledge with significantly less forgetting than standard transformers.
> >
> > These show promising signals that the design here could be scaled to more realistic settings and problems. Regarding your comment that parametric memory may reduce the flexibility of knowledge update and degrade other capabilities, our view is that these are all interesting and important research problems to address. Arguably, humans mostly learn through updating parametric memory, and we have no problem frequently updating our knowledge without hurting other abilities. There is also existing and ongoing research on better parametric memory designs that shows promising signals in alleviating these issues (e.g., the study of Lin et al. mentioned above).
> >
> > A concrete plan to examine the effect of our proposed designs on the broader capabilities obtained via regular language modelling could be to “interpolate” regular designs and our designs, and monitor the performance on both regular capabilities and those requiring deeper conceptual integration (e.g., reversal and related tasks). More concretely, for JEPA, this could be done via auxiliary losses as discussed earlier, where we mix concept-level prediction with regular token-level prediction with some controllable weighting factor (this is also the idea of the recent LLM-JEPA work by Huang et al.). For memory layers, a good plan is to replace standard MLP blocks with memory layers of varying levels of progressiveness (akin to the ablation studies in Berges et al.). These could enable us to examine how our proposed designs influence the regular broader capabilities (and vice versa), and how best to make them “co-exist” and (ideally) mutually benefit from each other.
> >
> > **4. Statistical Rigor**
> >
> > From the large set of experiments we ran throughout the study, the results are quite stable. For completeness, we ran the JEPA experiments in Figure 3 (Left) with 2 additional random seeds, and report the mean and standard deviation below:
> >
> > | | **#Rec = 1** | **#Rec = 2** | **#Rec = 3** |
> > | :--- | :--- | :--- | :--- |
> > | **#Sem/#Rec = 1** | $0.864 \pm 0.008$ | $0.814 \pm 0.003$ | $0.533 \pm 0.004$ |
> > | **#Sem/#Rec = 3** | $0.594 \pm 0.006$ | $0.565 \pm 0.006$ | $0.371 \pm 0.009$ |
> > | **#Sem/#Rec = 6** | $0.573 \pm 0.003$ | $0.402 \pm 0.012$ | $0.230 \pm 0.004$ |
> >
> > **5. Minor Questions**
> > - **$\mathcal{E}_A$ in Table 1**: This refers to the set of entities used for training (learning/inducing the reversal rules), as defined in Section 2 (Line 140).
> > - **"Fail entirely"**: We mean literally 0% accuracy. Across all baselines (depths 1–18, widths 768–1280, tied/untied embeddings), standard transformers achieve 0.0% on the reversal test set. We will make this explicit in the final revision.
> > - **Standard Transformer + Memory Layers**: We did experiment with this configuration, and it also yields 0% accuracy. We previously omitted this result for narrative flow, but will include it in the final version. This result confirms that addressing both components is necessary: while Memory Layers mitigate entanglement, without using JEPA-style objectives to bind the subject/object representations under surface-level predictions, the model still fails.
> >
> > We thank the reviewer again for the thoughtful comments and constructive suggestions. We will incorporate the improvements and additional results into the revised draft over the next few days. We look forward to addressing any additional questions or comments.
> >
> > **References**
> > - Berglund et al. The Reversal Curse: LLMs trained on "A is B" fail to learn "B is A". ICLR-24.
> > - Berges et al. Memory Layers at Scale. arXiv-24.
> > - Lin et al. Continual Learning via Sparse Memory Finetuning. arXiv-25.
> > - Lee et al. Difference Target Propagation. ECML-15.
> > - Barrault et al. Large Concept Models: Language Modeling in a Sentence Representation Space. arXiv-24.
> > - Huang et al. LLM-JEPA: Large Language Models Meet Joint Embedding Predictive Architectures. arXiv-25.

---

### Official Review · Reviewer_fjq7 · 2025-11-01

**Soundness:** 3
**Presentation:** 3
**Contribution:** 2
**Rating:** 4
**Confidence:** 3

**Summary:**

The authors demonstrate that JEPA-style architectures do not suffer from the same reversal curse, where transformers are not able to associate the latter parts of the sentence with earlier parts of the sentence and create the right associations (learning A=>B, but not B=>A). They further validate this finding by constructing a toy dataset of entity sets and relations and their reverses. Although an interesting finding (albeit intuitive because JEPA style architectures use the InfoNCE loss over different segments of the sentence, thereby creating an invariant representation bidirectionally), I am not sure that this capability or characteristic alone compels or convinces us the JEPA is better than transformers, given that the JEPA framework would require us to know the concept space a priori.

**Strengths:**

- The paper validates the existence and how the JEPA-style architecture solves this problem with a simple yet clear toy dataset.
- The authors also show that solving this reversal curse is also enables parametric forward-chaining and competitive performance on large arithmetic trees, which is a more real-world example of why this could matter.

**Weaknesses:**

- As the authors note, the JEPA setup requires one to know where the concept states live and the memory layer trick presumes unique name equals to unique concept.
- The dataset is very simplistic and toy-ish. We do not necessary want the models to learn a reverse map of everything (for example, A => B, does not actually imply B => A), and it is unclear to me why the reversal curse is a problem that is necessary to solve. Perhaps, a higher level motivation (beyond it's an important behavior study in cognitive science etc) would be helpful.

**Questions:**

- The memory layer idea is interesting, and it is reminiscent of the Memory Layers at Scale [1]. I wonder how the transformers would fare with a similar memory layer in Figure 5.

[1] Berges, Vincent-Pierre, et al. "Memory layers at scale." arXiv preprint arXiv:2412.09764 (2024).

---

> ### Author Response · Authors · 2025-11-21
>
> We thank the reviewer for the thoughtful assessment, and for recognizing that our work 1) validates the existence of the Reversal Curse while providing a working solution via JEPA-style architectures, and 2) demonstrates the usefulness of the reversal skill in enabling parametric forward chaining for solving complex arithmetic reasoning problems. We address the concerns/questions below.
>
> **1. On "Scaffolding" and Prior Knowledge**
> - As discussed in Section 6, we acknowledge that our current solution relies on a priori knowledge of concept locations and name-concept correspondence. However, we emphasize that such scaffolding was a deliberate experimental choice to ensure a rigorous, controlled scientific study. Our primary goal in this work is to isolate the *binding mechanism* as the root cause of the Reversal Curse. Through these controls, we provide a rigorous "existence proof" that when the different aspects of conceptual binding are addressed, the Reversal Curse can be broken.
> - While our experiments rely on such scaffolding, JEPA-style architectures do not inherently require it. We view our work as a foundational step that motivates future research on designs that could enable models to better learn/induce such bindings automatically from data, and also further exploration of the potential of JEPA-style architectures for language-related domains (as exemplified by recent work such as Huang et al., 2025).
>
> **2. Motivation: Why solve Reversal?**
> - The reviewer asks if solving the Reversal Curse is necessary, noting that “we do not necessarily want the models to learn a reverse map of everything”. We agree; however, we note that our goal is to *enable models to learn reversible associations when the underlying relation is indeed reversible* (as indicated by the large volume of bidirectional evidence in the training data).
> - The value of reversal also extends far beyond simple fact retrieval, as demonstrated in Section 4. More generally, consider a scenario where a model has internalized the semantics of $x$ and $y$, and witnesses the fact $f(x,y)=z$. With the ability to reverse, the model can now bind the deduced value $f(x,y)$ to the variable name $z$ within its parametric memory, which allows further deductions based on the semantics of $z$. While our arithmetic reasoning task (Section 4) is a simple instantiation of this, this skill is generalizable and could enable models to solve broader reasoning problems that remain challenging for standard Transformers, such as variable tracking in large codebases.
>
> **3. Standard Transformers with Memory Layers**
> - We *are* actually adopting the memory layer design in Berges et al., as mentioned in lines 361-362. We have also done experiments on standard Transformers with memory layers, and found that they do not exhibit generalization. These results were initially omitted to maintain narrative flow, but we will include a discussion of them in the next version. These results are also consistent with our analysis in Section 3: even with memory layers that mitigate representational entanglement, the issue of *inconsistency* (failure to bind representations of the same underlying concept when it switches roles across contexts) persists in standard Transformers. The synergy of operating in the concept space and disentangling representations is needed to achieve strong generalization.
>
> We thank the reviewer again for the thoughtful comments and constructive suggestions. We will incorporate the improvements into the revised draft over the next few days. We look forward to addressing any additional questions or comments.
>
> **References**
> - Huang et al. LLM-JEPA: Large Language Models Meet Joint Embedding Predictive Architectures. arXiv-25.
> - Berges et al. Memory Layers at Scale. arXiv-24.

---

### Author Response · Authors · 2025-11-26
**Manuscript Revision**

We thank all reviewers for the thoughtful and constructive feedback. We have carefully revised the manuscript to address the concerns raised across the reviews and incorporate the suggested improvements. In particular, the revision:
- Expands the discussion around the JEPA architecture (LeCun, 2022) in Appendix B, including its relationship to our JEPA formulation and terminology (**Reviewer NLXN**), as well as additional stability results (**Reviewer 4J4R**).
- Adds previously omitted discussions, most notably the standard transformer + memory layer ablation (around line 374) (**Reviewers fjq7, 4J4R**).
- Improves the clarity of empirical results & explanations, and consistency in terminology throughout the paper (**Reviewers fjq7, 4J4R, 3p1K, NLXN**).

We believe these strengthen the paper’s clarity, empirical rigor, and broader relevance. We look forward to addressing any further questions or comments.

---

### Meta-Review · Area_Chair_HaN7 · 2025-12-17

**Summary:**

The reviewers questioned the practical applicability and generalization of the proposed method, noting that the experiments relied heavily on simplistic toy datasets. They expressed concern that the framework requires a priori knowledge of the concept space, which is often unavailable in real-world scenarios, and doubted whether the method would scale to realistic natural language processing tasks or coexist with capabilities like in-context learning. There was also skepticism regarding the necessity of solving the reversal curse, with one reviewer noting that logical implications are not always bidirectional in natural language.

Technically, the reviewers found the architectural choices under-justified, asking why JEPA and the specific top-k softmax memory layer were chosen over simpler alternatives or standard transformers with memory augmentation. They criticized the lack of rigorous baselines, such as a transformer with the memory layer but without JEPA, and the absence of statistical robustness measures like error bars and multiple random seeds. Finally, several reviewers found the explanation of the JEPA implementation unclear and requested better definitions of how the recognition and semantic modules map to the transformer architecture.

**Reviewer Concerns:**

The authors provided some reasonable justifications for their choice of datasets and model designs. They also worked to improve clarity and added some additional data to address concerns arounf the statistical robustness. I do not think they fully addressed the concerns about scaling, though.

**Reviewer Scores:**

The original reviewer scores were 4,6,6,4. The final reviewer explicitly said they would *not* raise their score, but my guess is that at least two reviewers may have increased theie score, so we may have had final scores along the lines of 5,6,7,4. This would be a borderline accept.

---

### Decision · Program_Chairs · 2026-01-26

Accept (Poster)